# Assimilation of GNSS Zenith Delays and Tropospheric Gradients: A Sensitivity Study utilizing sparse and dense station networks

Rohith Thundathil[1,2], Florian Zus[2], Galina Dick[2], Jens Wickert[1,2]

[1]Technische Universität, Berlin, Germany

[2] GFZ Helmholtz Centre for Geosciences, Potsdam, Germany

*Correspondence to*: Rohith Thundathil (r.thundathil@tu-berlin.de)

**Abstract.** The assimilation of Global Navigation Satellite System (GNSS) zenith total delays (ZTDs) into numerical weather models improves weather forecasts. In addition, the GNSS tropospheric gradient (TG) estimates provide valuable insight into the moisture distribution in the lower troposphere. In this study, we utilize a newly developed forward operator for TGs

to investigate the sensitivity effects of incorporating TGs into the Weather Research and Forecasting model at varying station network densities. We assimilated ZTD and TGs from sparse and dense station networks (0.5 and 1-degree). Through this study, we found that the improvement in the humidity field with the assimilation of ZTD and TGs from the sparse station network (1-degree resolution) is comparable to the improvement achieved by assimilating ZTD only from the dense station network (0.5-degree resolution). These results encourage the assimilation of TGs alongside ZTDs in operational

weather forecasting agencies, especially in regions with few GNSS stations. Conversely, assimilating TGs alongside ZTDs from sparse GNSS networks can be a cost-effective way to enhance the accuracy of the model fields and subsequent forecast quality.

## 1 Introduction

Global Navigation Satellite Systems (GNSS) have become integral to our everyday lives. It significantly revolutionized how

we determine our position, navigate, and keep track of time. The most profound application of GNSS has been in civilian and commercial uses, such as positioning, navigation, and timing. However, GNSS is increasingly valuable for geosciences in accurately sensing atmospheric and surface properties and other geophysical parameters. Additionally, it can be used to derive the Earth's surface properties, deformation, and other geophysical parameters (Wickert et al., 2020).

Monitoring atmospheric water vapor with GNSS regional ground networks has helped bridge gaps in established

meteorological observing systems. GNSS is distinguished from other observation systems by its numerous benefits, such as low operating costs, all-weather availability, and exceptional spatio-temporal resolution. The total number of GNSS stations worldwide exceeds 10,000. European networks, with about 3,000 stations, enhance regional weather forecasts. Incorporating advanced GNSS-based observations allows us to provide high-quality information with high spatio-temporal distribution in

operational weather forecasting models worldwide. This is essential for accurately modeling the atmosphere, especially for

predicting heavy precipitation and severe weather events, which are significant challenges in weather research.

Since 1992, GNSS signals have been utilized to monitor the atmosphere through ground-based stations ('GNSS meteorology'). GNSS meteorology uses the time delay of radio signals traveling from the satellite to the station to monitor atmospheric water vapor. Zenith Total Delay (ZTD) is a key measurement in GNSS meteorology (Bevis et al., 1992), closely linked to the Integrated Water Vapor (IWV) above the station. ZTD data are available in near-real-time (NRT) from several

European station networks, such as the European Meteorological Network Global Navigation Satellite Systems Water Vapor Program (EGVAP). Once adjusted for ionospheric effects, the delay caused by the troposphere in transmitting GNSS signals between satellites and stations is estimated. The ZTD has been utilized by various operational forecast agencies. Several assimilation studies have been performed with ZTDs and found that they enhance the accuracy of the forecasts. For example, Vedel and Huang (2004) showed that the ZTD assimilation improved the prediction of strong precipitation. Poli et al. (2007)

also found a positive impact on the prediction of short-term precipitation and quantitative precipitation forecast scores for total precipitation over France between +12 and +36 hours after analysis time. The Action de Recherche Petite Échelle Grande Échelle (ARPEGE) global model was used here to understand the assimilation impact of synoptic-scale circulations and precipitation forecasting during spring and summer. Yan et al. (2009) performed assimilation experiments using the Aire Limitée Adaptation Dynamique Développement International (ALADIN) model. They found that assimilating ZTDs

improved the meso-nonhydrostatic precipitation forecasts for a heavy rainfall event over the Mediterranean region. Boniface et al. (2009) assimilated GNSS data into the Applications of Research to Operations at Mesoscale (AROME) model. They showed improvement in predicting the spatial extent of the precipitation. Lindskog et al. (2017) used the HIRLAM–ALADIN (High Resolution Limited Area Model; Aire Limitée Adaptation Dynamique Développement International) Research on Mesoscale Operational NWP in Euromed (HARMONIE) Applications of Research to Operations at Mesoscale

(HARMONIE–AROME) model to test ZTD data assimilation. Their findings show that including ZTD as an additional observation type enhances forecast accuracy, emphasizing the possibility of enhancing data assimilation by combining GNSS ZTD with other observations. Rohm et al. (2019) conducted assimilation studies using the Weather Research and Forecast (WRF) model ZTD operator. They found that the ZTD assimilation altered the moisture field and precipitation rather than other parameters, such as the pressure or temperature field. GNSS observations enhance forecasts within 24

55    hours, with the most impact at a 9-hour lead time. Giannaros et al. (2020) and Caldas-Alvarez and Khodayar (2020) also demonstrated the significant benefits of incorporating GNSS ZTD data to improve precipitation and water vapor forecasts. Their studies used the WRF model in a broader Mediterranean region and the COSMO-CLM (COnsortium for Small-scale MOdeling in CLimate Mode) model in the central European region, respectively. Lagasio et al. (2019) discovered that integrating diverse Sentinel-1 and GNSS ZTD observations into the WRF model provides significant advantages for

forecasts, offering detailed information on the wind field and water vapor content. Singh et al. (2019) found that using ZTD observations from a ground-based GNSS network improved humidity, air temperature, and wind forecasts in the Indian region. Assimilating these observations reduced forecast errors in wind fields and enhanced rainfall predictions to some

extent. Mascitelli et al. (2019, 2021) successfully utilized the Regional Atmospheric Modeling System at the Institute of Atmospheric Sciences and Climate (RAMS@ISAC) model to incorporate GNSS ZTD data, leading to a significant

enhancement in short-term water vapor prediction with minimal impact on precipitation forecasts. Yang et al. (2020) found that combining ZTD and radar data improved the accuracy of heavy rainfall location and intensity. They also discovered that using a broader horizontal localization scale instead of the convective scale for radar data assimilation enhanced the impact of ZTD data. Risanto et al. (2021) found that assimilating Global Positioning System (GPS) precipitable water vapor improved short-range North American monsoon precipitation forecasts by reducing errors and biases in the initial conditions

of the weather model. This enhanced the model's ability to capture nocturnal convection of mesoscale convective systems and improved precipitation timing.

ZTDs are the only GNSS-derived moisture data used operationally; however, they provide limited atmospheric information. New observations must augment the existing observations, providing additional information. According to Bennitt & Jupp (2012) and Mahfouf et al. (2015), the limitations of ZTD lie in its inability to provide information on horizontal or vertical

atmospheric gradients. Tropospheric gradient (TG) is another variable derived from the GNSS (Bar-Sever et al., 1998). In simple terms, TGs mainly provide information on the moisture's change (or "gradient") in a specific direction. Bar-Sever et al. (1998) showed that including TGs in GPS geodesy enhances accuracy and precision, with the estimated gradients matching real atmospheric moisture patterns observed by a water vapor radiometer (WVR). Walpersdorf et al. (2001) used the ALADIN model to validate GPS TGs at five stations. Iwabuchi et al. (2003) found a strong correlation between these

gradients and moisture fields, with TGs typically pointing from dry to moist regions. Brenot et al. (2013) observed similar phenomena in their deep convection studies. Li et al. (2015) showed that better observation geometry improves gradient estimation accuracy. Morel et al. (2015) analyzed data from 12 Corsican stations using different software. Douša et al. (2016) analyzed data from hundreds of stations in central Europe and confirmed that GNSS TGs reflect real tropospheric features. Kačmařík et al. (2019) highlighted the sensitivity of TGs to processing options, emphasizing that real-time accuracy

depends on high-quality satellite data.

Thundathil et al. (2024) illustrate the operator implementation and assimilation of TGs in the WRF model. The TG operator (Zus et al., 2023) was incorporated into the WRF data assimilation (WRFDA) system in version 4.4.1. The source codes are published online for the research community worldwide. The study accomplished a two-month assimilation impact study to obtain statistical confidence on the impact focused on Europe. The observations for the impact studies were collected from

the GFZ Helmholtz Centre for Geosciences. The study quantified the impact, showing promising improvements by adding TGs on top of ZTDs. In this study, we aim further to investigate the potential of TGs through a sensitivity experiment. We wish to analyze under which circumstances TGs provide information when combined with ZTDs to improve the initial conditions for numerical weather prediction.

## 2. GNSS ZTD and Tropospheric Gradients

The tropospheric delay is caused by the signal traveling through the neutral atmosphere. It is parameterized in the GNSS analysis with mapping functions (MFs), zenith delay, and gradient terms. The tropospheric delay $T$ at the station is expressed as a function of the elevation angle $e$ and the azimuth angle $a$:

$$T(e,a) = m_h(e).Z_h + m_w(e).Z_w + m_g(e)[\cos(a).N + \sin(a).E] \tag{1}$$

where $Z_h$ is the zenith hydrostatic delay (ZHD), $Z_w$ is the zenith wet delay (ZWD), and $N$ and $E$ are the north and east gradient components. The hydrostatic, wet, and gradient MFs are denoted $m_h$, $m_w$, and $m_g$, respectively. The ZTD, $Z$, is given by

$$Z = Z_h + Z_w. \tag{2}$$

The forward operator for the ZTD, along with the tangent linear and adjoint operators, is already integrated into the WRFDA system. The ZTD is calculated through:

$$Z = 10^{-6} \int \Psi \, dz \tag{3}$$

where the refractivity, $\Psi$, is a function of pressure, temperature, and humidity (Thayer, 1974), with $z$ denoting the height above the station. In the GNSS data analysis, the ZTD ($Z$), the north gradient component ($N$), and the east gradient component ($E$) are estimated with geodetic parameters through least square adjustment (Gendt et al., 2004). The three quantities, depending on the state of the atmosphere in the vicinity of the station, are considered observations. The TG forward operator uses a fast approach which works as follows: for the given station location, we utilize a closed-form expression that depends on the north–south and east–west horizontal gradients of refractivity (as outlined in Davis et al., 1993). This enables the calculation of the north and east gradient components through

$$N = 10^{-6} \int z\Psi_y dz \tag{4}$$

$$E = 10^{-6} \int z\Psi_x dz \tag{5}$$

Here, $x$, $y$, and $z$ represent the Cartesian coordinates and partial derivatives are denoted by the corresponding subscripts. Similar to the computation of ZTDs, the TGs are also calculated using numerical integration.

Recently, Zus et al. (2023), developed the TG operator, which has been implemented into the WRFDA system version 4.4.1. Initial DA experiments conducted for the dense GNSS station network in Germany have shown promising results (Thundathil et al., 2024).

## 3. Model setup

In this study, the WRF model version 4.4.1 is used with the Advanced Research WRF (ARW) core (Skamarock et al., 2008). WRF has been widely used for research within a large community and also serves as a model for operational forecasting at various agencies worldwide (Powers et al., 2017).

The model domain was configured with a 0.1-degree (approx. 11 km) horizontal resolution and 250 x 250 grid points. The number of vertical levels in the model is 50, extending from Earth's surface to an altitude of 50 hPa. The initial and boundary conditions were obtained from the European Centre for Medium-Range Weather Forecasts operational analysis, which had a spatial resolution of 0.14 degrees (approx. 16 km). In this sensitivity study, we are using the GNSS observations from the Benchmark data set which was collected within the European COST Action ES1206 GNSS4SWEC (Advanced

GNSS tropospheric products for monitoring severe weather and climate; Douša, Jan, et al., 2016). The GNSS stations in central Europe covering Germany, the Czech Republic, and part of Poland and Austria provided the data during this campaign. Figure 1 shows the WRF model domain with the GNSS stations.

The WRF model physics settings are the same as those in Thundathil et al. (2024). The radiation parameterization scheme used in this study is based on the Rapid Radiative Transfer Model for General Circulation Models (RRTMG) developed by

130 Iacono et al. (2008). This model is recognized for its accuracy and efficiency in calculating long-wave and short-wave fluxes and heating rates, making it particularly suitable for applications in general circulation models.

For the cloud microphysics, we implemented the Thompson double-moment scheme (Thompson et al., 2008), which can predict mixing ratios for cloud water, rain, ice, snow, and graupel. The planetary boundary layer scheme utilized in this simulation is the Yonsei University (YSU) scheme (Hong et al., 2010; Hong & Lim, 2006). The YSU is a non-local scheme

with first-order closure that incorporates counter-gradient and explicit entrainment terms into the turbulence flux equation.

This study also employed the unified Noah land surface model (Chen & Dudhia, 2001). This model consists of four layers and is designed to predict soil temperature and moisture, canopy moisture, and snow cover. It takes into account various factors, including root zone dynamics, evapotranspiration, soil drainage, runoff, vegetation categories, and soil texture. This comprehensive approach yields valuable information on sensible and latent heat fluxes related to the boundary layer,

including an enhanced treatment for urban areas.

To simulate the model accurately at a non-convective-scale resolution, it is crucial to include convection parameterization, which helps represent the statistical effects of sub-grid-scale convective clouds. For this purpose, we used the Grell–Freitas ensemble scheme (Grell & Freitas, 2014), which integrates a probability density function with data assimilation techniques.

### 3.1 Data

For the assimilation experiment, we had GNSS tropospheric products from 430 stations which belong to the core of the Benchmark data set which was collected within the European COST Action ES1206 GNSS4SWEC (Douša, Jan, et al., 2016). The GNSS ZTDs and TGs were obtained in precise point positioning mode utilizing the G-Nut/Tefnut software

(Václavovic et al., 2014). Details on the quality of the tropospheric products can be found in Kačmařík et al. (2019). To ensure a homogeneous set of observations across the domain, we excluded collocated and clustered stations and specifically chose GNSS stations with data availability exceeding 75%. In addition, to complying with our WRF model domain, we carried out a simple thinning of observations ('homogenization' of station distribution). The thinning method was conducted in two steps. First, a 0.5-degree mesh was constructed. Then, GNSS stations were selected based on their proximity to the mesh-grid point. Finally, we obtained a station network with a resolution of about 0.5 degrees. After these steps, we were left with around 250 GNSS stations over the Benchmark domain. For the sensitivity experiment, we created another thinned station network with a resolution of about 1 degree that contained around 110 stations (see Fig. 1). The same thinning procedure was used again. In order to compare the simulations with respect to independent GNSS observations, we removed around 18 stations in Germany from the total dataset. In line with the approach of Thundathil et al. (2024), we intentionally excluded 18 stations from our dataset for validation purposes. These excluded stations were chosen strategically to maintain a balanced spatial distribution, aligning with the locations of the German Weather Service (DWD: Deutscher Wetterdienst) radar stations. The remaining stations included in the model are referred to as "allowed" stations. This method enabled us to analyze improvements with respect to independent observations. We utilized analyses from our control experiment to implement a "fixed" bias correction, addressing potential biases in the GNSS dataset (Thundathil et al., 2024). The two months of simulation from the control experiment were employed to perform a station-specific bias correction for the GNSS ZTDs and TGs. The standard observation error for ZTDs in operational forecasting typically ranges from 5 to 15 mm. Similar to our previous study, Thundathil et al. (2024), the same observation errors were adopted: 8 mm for the ZTDs and 0.65 mm for the TGs. Given the high quality of the observations from the Benchmark campaign dataset, we have maintained these same error values in this study. The North and East gradient observation errors were calculated based on an analysis of the observation-minus-background (OB) statistics from the control run. OB statistics encompass both observation and model errors. An observation error of 0.65 mm was conservative since we did not want to force the model too much to the observations. To improve the analysis, we assimilated a set of conventional observations in addition to the GNSS observations. The conventional observations included a network of SYNOP stations across Europe. Radiosonde measurements offered a detailed view of the atmospheric thermodynamic structure at launch points. In order to maintain simplicity within the DA system, we limited conventional datasets to SYNOP surface observations and radiosondes. The number of observations ranges from 1029 to 1225 for SYNOP stations and 4 to 35 for radiosondes.

**3.2 DA Framework**

In this study, we used the deterministic three-dimensional variational (3DVAR) DA system. It uses an iterative minimization of the cost function $J$ with a background constraint and an observation constraint. The 3DVAR cost function equation is given by

$$J(x) = \frac{1}{2}(x - x_b)^T \mathbf{B}^{-1}(x - x_b) + \frac{1}{2}(y - \mathbf{H}(x))^T \mathbf{R}^{-1}(y - \mathbf{H}(x)) \tag{6}$$

The variables $x$, $x_b$, and $y$ are column vectors that represent the model state, the background (or first guess), and the observation state, respectively. The forward operator, denoted by **H**, maps the model state vector to the observation vector. **B** represents the background error covariance matrix, while **R** represents the observation error covariance matrix. The observations are assumed to be uncorrelated, so **R** is a diagonal matrix. **B** is a square, positive, semi-definite, and symmetric matrix that contains the variances of the background forecast errors along the diagonal and their covariances in the upper and lower triangles of the matrix. We computed a climatological background error covariance matrix using the National Meteorological Center (NMC) method (Parrish & Derber, 1992). The NMC method involves calculating forecast difference statistics to obtain the forecast error covariance. The *B* matrix for the regional simulations was derived from the forecast statistics by analyzing differences in 24-hour and 12-hour predictions over a month using data from May 2013. We chose the CV5 option for independent control of moisture levels, as it minimizes interference from other control variables. CV5 refers to a version of the background error covariance matrix used in the WRF model. It incorporates five control variables: stream function ($\Psi$), unbalanced velocity potential ($\chi_u$), unbalanced temperature ($T_u$), pseudo-relative humidity ($RH_s$), and unbalanced surface pressure ($P_{s,u}$). Pseudo-relative humidity is the ratio of $Q$ to $Q_{b,s}$, where $Q_{b,s}$ represents the saturated specific humidity of the background field.

The assimilation system used a Six-hourly DA cycle framework, with assimilations over May and June 2013. Spin-up is essential for the model to stabilize with the initial and boundary conditions, enabling it to respond accurately to any desired inputs. Only after a sufficient spin-up period can the model forecasts be considered reliable for further analysis through data assimilation (DA). For our study, we adopted a 12-hour spin-up before the assimilation (Lauer et al., 2023). We conducted two sets of experiments. The first set comprised three experiments: 1) Control run with assimilation of conventional data only, 2) ZTD run assimilating ZTDs on top of the Control run, and 3) ZTDGRA run assimilating ZTDs and TGs on top of the Control run. We term the second and third experiments ZTD_0.5° and ZTDGRA_0.5° to distinguish them from the second set of experiments, making them easier for readers to understand. In the RUC, an hourly forecast output was generated after each assimilation cycle for the next five hours, which resulted in one analysis and five forecasts.

The second set of experiments was performed to analyze the sensitivity of the gradient observations by de-densification of the GNSS stations. We de-densified the GNSS stations from a roughly 0.5-degree to a 1-degree station network and then performed the assimilation experiments. Hence, ZTD_1.0° and ZTDGRA_1.0° runs were conducted similarly to ZTD_0.5° and ZTDGRA_0.5°, respectively, but with the assimilation of observations from the 1-degree station network. The assimilation cycle starts from 5 May 2013 00 UTC to 29 June 2013 18 UTC, the entire available data timeline from the benchmark campaign. The DA framework of the experiments is shown in Figure 2.

## 4. Results

### 4.1 Impact of GNSS data

To evaluate the impact of assimilating TGs on top of ZTDs, we conducted a comparative analysis of the results from two-month-long assimilation experiments using data from GNSS stations. Specifically, we compared the analyses and forecasts obtained from these experiments against observations from GNSS stations, both assimilated and independent stations (which were not assimilated). The assimilated stations are termed as "allowed stations", and the independent stations are termed as "excluded station," similar to Thundathil et al. (2024). The quantitative comparison involved hourly GNSS station data,

which were assessed against six-hourly data assimilation (DA) analyses and five-hour forecasts initialized from these analyses. The model simulation for two months in each experiment comprises six-hourly analyses and a five-hour forecast in between two DA cycles. Hence, the model simulations consist of hourly model outputs of analyses and forecasts. This hourly model output is compared to the corresponding GNSS station data for each experiment to calculate the root mean square error (RMSE). The model ZTDs and TGs at the locations of each specific station are computed for the RMSE. We

term this the "station-specific RMSE." This section focuses on comparing the first set of experiments, labeled ZTD_0.5° and ZTDGRA_0.5°, with the control run.

Figure 3 is simplified in a percentage analysis of the station-specific RMSE plot (please refer to Figures A1-A6 in the appendices) for a more straightforward interpretation. Here, the control experiment was kept as the base experiment, and the ZTDs and ZTDs plus TGs assimilation experiments in dense and sparse configurations were compared. A reduction in the

225 RMSEs indicates improvement in the assimilation experiment. The reduction in RMSE is represented as a percentage increase, which means the higher the reduction, the higher the percentage. Table 1 lists the average of the station-specific RMSEs of all the DA experiments for the two months. Figure 3 (top-left) demonstrate that the ZTDGRA_0.5° experiment yielded the lowest mean RMSE values for the ZTD parameter among all runs with the highest percentage reduction of 43 % in the mean station-specific RMSE with respect to the control run. This indicates the successful impact of gradient

assimilation. Specifically, referring to Table 1, the mean RMSE for the ZTD variable decreased from 14.4 mm in the control run to 8.3 mm in the ZTD_0.5° run, and further to 8.2 mm in the ZTDGRA_0.5° run. Improvements were observed not only in ZTD values but also in the gradient components. Both the north and east gradient components exhibited reductions in RMSE. For the north gradient, RMSE decreased from 0.62 mm in the control run to 0.52 mm in the ZTD_0.5° run, and further to 0.49 mm in the ZTDGRA_0.5° run. Similarly, for the east gradient, RMSE decreased from 0.66 mm in the control

run to 0.54 mm in the ZTD_0.5° run, and then to 0.50 mm in the ZTDGRA_0.5° run. These reductions in RMSE values underscore the significant improvements achieved by assimilating TGs, which enhanced the moisture field representation in the model state. The findings highlight the synergistic relationship between ZTDs and TGs assimilation, where assimilating ZTDs contributes to the refinement of TG components, and vice versa.

To confirm that these improvements were not solely due to comparisons with observations from the assimilated GNSS

stations, we extended the analysis to include 18 independent GNSS stations that were excluded from the assimilation

process. Again the highest RMSE reduction of 42 % was observed in the ZTDGRA_0.5° run compared to the control run. The RMSE for the ZTD variable decreased from 13.7 mm in the control run to 8.2 mm in the ZTD_0.5° run, and further to 8.0 mm in the ZTDGRA_0.5° run (Table 1). Similar trends were observed for the gradient components. For the north gradient, RMSE decreased from 0.59 mm in the control run to 0.49 mm in the ZTD_0.5° run and 0.47 mm in the ZTDGRA_0.5° run. For the east gradient, RMSE reduced from 0.63 mm in the control run to 0.51 mm in the ZTD_0.5° run, and then to 0.48 mm in the ZTDGRA_0.5° run.

These consistent results across both assimilated and independent GNSS station data demonstrate the robust improvements achieved through gradient assimilation on top of ZTDs. The two-month-long statistical evaluation confirms that the combined assimilation of ZTDs and TGs improves the humidity field.

To quantify the relative impact of GNSS observations compared to other point observations in the study, specifically the SYNOP station data, we utilized the Desroziers method. Desroziers method is an effective diagnostic tool used to evaluate the impact of various observations. By analyzing the Innovation (Observation minus Background, OMB) and Residual (Observation minus Analysis, OMA) statistics, we can estimate the covariances of observation and background errors. This analysis helps us determine the relative influence of different types of observations on the overall analysis.

The relative impact of an observation is determined by the ratio of the estimated observation error covariance R to the estimated background error covariance B. The respective error covariances are calculated as below:

$$\boldsymbol{R} = E[(y - Hx^a)(y - Hx^a)^T] \tag{7}$$

$$\boldsymbol{B} = E[(y - Hx^b)(y - Hx^a)^T] \tag{8}$$

Here $(y - Hx^b)$ is the innovation and $(y - Hx^a)$ is the residual where $x^a$, $x^b$, and $y$ are the model state vectors for analysis, background and observations, respectively.

The higher the value of the ratio $\boldsymbol{B}/\boldsymbol{R}$, the higher the impact of the observation. The observations likely to enhance the model or lead to effective assimilation fall within the range of 0.5 to 3. A value below 0.5 suggests that the observation has large error, making it unreliable for assimilation. Conversely, values above 3 indicate that the observation forces the background towards the observation, which may result from a small observation error or a bias in the observation. After analyzing 220 DA cycles, the average $\boldsymbol{B}/\boldsymbol{R}$ ratios were as follows: SYNOP at 1.4, ZTD at 2.8, NG at 1.8, and EG at 1.5. These values indicate that the impacts of the observations are well within the acceptable range. Additionally, the TGs, have an impact in the assimilation system. The North and East gradient values indicate that the assimilation was effective. The ZTD observation has higher values, which might indicate that the observation error assigned to ZTDs could be higher than the current observation error value of 8 mm.

## 4.2 Sensitivity analysis

A sensitivity experiment was conducted to better understand the conditions under which the assimilation of TGs, in addition to ZTDs, improves the representation of the humidity field. For this purpose, a second set of experiments was designed using

GNSS data assimilation from a sparser 1-degree network. This allowed for a focused analysis of the additional impact brought by gradient assimilation. As with the dense network experiments, two configurations were tested: a ZTD assimilation run (ZTD_1.0°) and a combined assimilation run incorporating both ZTDs and TGs (ZTDGRA_1.0°).

When comparing data from stations included in the assimilation process, the ZTDGRA_1.0° experiment exhibited the lowest mean RMSE values for ZTD, similar to the results observed with the dense network configuration. The highest reduction was 42% for the ZTDGRA_1.0° run. Specifically, referring to Table 1, the ZTD variable's mean RMSE decreased from 14.4 mm in the control run to 9.2 mm in the ZTD_1.0° run and further to 8.7 mm in the ZTDGRA_1.0° run. For the TGs, RMSE values showed improvements in both the north and east components. The north gradient RMSE reduced from 0.63 mm in the control run to 0.55 mm in the ZTD_1.0° run and to 0.51 mm in the ZTDGRA_1.0° run. Similarly, the east gradient RMSE decreased from 0.66 mm in the control run to 0.58 mm in the ZTD_1.0° run and further to 0.52 mm in the ZTDGRA_1.0° run.

A comparable trend was observed with data from 18 independent GNSS stations excluded from the assimilation. For these stations, the ZTD variable's mean RMSE decreased from 13.7 mm in the control run to 9.0 mm in the ZTD_1.0° run and further to 8.5 mm in the ZTDGRA_1.0° run, which was a reduction of 38 % in ZTDGRA_1.0° from 34% in the ZTD_1.0° run. The north gradient RMSE dropped from 0.59 mm in the control run to 0.52 mm in the ZTD_1.0° run and to 0.49 mm in the ZTDGRA_1.0° run (Table 1). Similarly, the east gradient RMSE declined from 0.63 mm in the control run to 0.55 mm in the ZTD_1.0° run and further to 0.51 mm in the ZTDGRA_1.0° run.

From the RMSE values, we conclude that, in particular, for a sparse network configuration, we can expect a significant impact on the assimilation of TGs on top of ZTDs. For example, suppose we utilize the RMSE of ZTDs for the independent stations as an indication of the improvement in the (integrated) water vapor field. In that case, referring to Table 1, the drop in the RMSE from 8.2 mm in the ZTD_0.5° experiment to 8.0 mm in the ZTDGRA_0.5° experiment is smaller than the drop from  9.0 mm in the ZTD_1.0° experiment to 8.5 mm in the ZTDGRA_1.0° experiment. A similar trend can be seen when we utilize the RMSE of ZTDs for the 'allowed' stations.

The most striking feature was that the RMSE reduction of the ZTDGRA_1.0° run was similar to the ZTD_0.5° run. In other words, the assimilation of ZTDs and TGs from a sparse station network performed equally well as that of only ZTDs from the dense station network. In order to illustrate this visually, the analysis increments of ZTDGRA_1.0° and ZTD_0.5° runs for consecutive DA cycles were analyzed. Figure 4 shows five analysis increments from the first DA cycle on 6 May 2013 00 UTC until 7 May 2013 00 UTC, with assimilation every six hours. Figure 4 is a zoomed-in map that covers only the countries where the GNSS stations were located. The rows in the plot refer to the corresponding DA cycles with ZTDGRA_1.0° on the left column and ZTD_0.5° on the right. DA cycle 1 refers to Figures 4a and 4f, and DA cycles two, three, four, and five refer to Figures 4b and 4g, Figures 4c and 4h, Figures 4d and 4i, and Figures 4e and 4j, respectively. The water vapor mixing ratio over the domain is vertically averaged for the first 16 model levels to portray the impact from the surface level up to the lower troposphere (approx. 6 km height). From the analysis increment comparison, a close match was observed between the two experiments with respect to respective assimilation cycles. From a visual inspection of the plots,

the sparse-network assimilation of ZTD and gradient run had the same structures as seen in the dense-network assimilation of the ZTD alone run. Quantitatively, the similarity of the ZTD_0.5° and ZTDGRA_1.0° runs at respective assimilation cycles can be computed by the structural similarity (SSIM) index parameter. The Structural Similarity Index (SSIM) is a metric used to quantify the similarity between two images. The value ranges from 0 to 1 where 0 shows no similarity and 1 shows a perfect similarity. Here is a short explanation of the computation of SSIM in our study:

$$SSIM(A,B) = \frac{(2\mu_A\mu_B + c_1)(2\sigma_{AB} + c_2)}{(\mu_A^2 + \mu_B^2 + c_1)(\sigma_A^2 + \sigma_B^2 + c_2)} \tag{9}$$

Here A and B represent the images in the left column (ZTDGRA_1.0°) and the right column (ZTD_0.5°) respectively. WVMR is the moisture variable presented here in the images with a span of 101 colors. $\mu_A$ and $\mu_B$ are the mean, $\sigma_A^2$ and $\sigma_B^2$ are the variance, and $\sigma_{AB}$ is the covariance. The variables $c_1$ and $c_2$ are computed based on the color span in the images.

$$c_1 = (k_1 L)^2 \tag{10}$$
$$c_2 = (k_2 L)^2 \tag{11}$$

Here $k_1$ and $k_2$ are 0.01 and 0.03 by default. $L$ here is the total number of colors in the color bar minus one. Hence the values of $c_1$ and $c_2$ comes to 1 and 3. Here, we computed the SSIM at all five assimilation cycles, showing a considerable similarity of the SSIM index greater than 0.98.

Finally, we took a closer look at the background and analyzed humidity profiles. In order to analyze the humidity profile correction in the assimilation experiment, we computed the RMSE of specific humidity profiles from model simulations with respect to ERA5 (Hersbach et al., 2020) at five locations spread equidistantly across the domain (for details, see Thundathil et al., 2024). The RMSE of the profiles with respect to ERA5 averaged over the two months is shown in Figure 5. There were 220 DA cycles, and with five profile RMSE comparisons at each cycle, the total number of profiles totaled 1100. From the figure, the RMSE of the ZTDGRA_1.0° run appears to overlap with the ZTD_0.5° run. This shows that the information passed into the model when TGs are assimilated on top of ZTDs for sparse network configurations is roughly as effective as the assimilation of ZTDs from the dense network configuration. This finding is particularly relevant for those aiming to densify their existing GNSS networks for weather prediction purposes. Before the costly installation and maintenance of additional (single or dual frequency) GNSS stations, she/he should consider the assimilation of TGs on top of the ZTDs.

### 4.3 Forecast impact

To understand how long the effects of GNSS observations assimilation persist within the model, we conducted simulations of 24-hour forecasts based on a three-day analysis. Each day included four assimilation cycles, resulting in a total of 12 forecasts, each covering 24 hours. The forecast is better validated with independent observations that are not assimilated into the model. With the 18 excluded GNSS stations, we can directly compare the model forecast with observations from the GNSS stations. Figure 6 compares the 12 forecast average with the GNSS ZTDs and TGs, including the North and East Gradients, to compute the standard deviation. We analyzed three impact experiments: ZTD_1.0°, ZTD_0.5°, and

ZTDGRA_1.0°, in addition to the control run. As anticipated, the effects of the three impact experiments gradually diminish and converge with the control run. If we define the endpoint of the impact as the moment when the standard deviation of the impact experiment aligns with that of the control run, then the duration of the impact is 12 hours. The effects of the assimilation last for no more than 12 hours, which is quite reasonable for moisture data assimilation. Additionally, it is important to note that incorporating TGs along with ZTDs enhances the forecast. Furthermore, the forecast impact of ZTDGRA_1.0° is comparable to that of ZTD_0.5°.

## 5. Conclusions

The TGs contain valuable information that has yet to be fully utilized by numerical weather models. From the assimilation experiments, we conclude that TGs, when assimilated in addition to ZTDs, enhance the accuracy of the humidity fields, thereby increasing the forecast accuracy. The work by Thundathil et al. (2024) already provided evidence that gradient observations positively impacted the analyses and forecast. The important result of this paper is the dependency of the impact of gradient observations on the network configuration. Since TGs can be roughly related to horizontal ZTD gradients, it was hypothesized that the impact of this new observation type would be beneficial, particularly for a sparse network configuration (Zus et al., 2019). Our results utilizing the state-of-the-art data assimilation system of WRF and GNSS tropospheric products from the Benchmark campaign prove this to be the case.

GNSS stations are available worldwide, but the station density varies from place to place. For example, the dense GNSS station network in Europe, with its near-real-time data provision capability, is already in its current status very effective in filling gaps in the humidity fields required for operational weather forecasting. However, in regions with a sparse GNSS station network or remote regions with isolated GNSS stations, the provided ZTD data leaves significant gaps in the highly variable humidity field. These gaps can be filled utilizing TGs.

NWMs will run globally at high resolution in the near future. For instance, ECMWF's global operational forecast already has a resolution of 9 km. In the future, we will also have convection-permitting scale resolution models running on a global scale, which would demand more observations for their initialization. We expect that the assimilation of GNSS TGs, in addition to ZTDs, helps to close gaps in the knowledge of the humidity field.

### Appendices

A detailed analysis of the assimilation impact of the GNSS data products is depicted through additional figures. The six figures and the table in this section provide supporting information on how Figure 3 in the main article was derived. The specific impact of the assimilation due to ZTDs and TGs with both dense and sparse assimilation setups are shown through the standard deviation compared to each GNSS station. The statistics were derived using the analysis and compared to the assimilated GNSS stations and independent GNSS stations, which were excluded from the assimilation dataset. We term the

assimilated stations "Allowed" and the independent stations "Excluded." Please refer the Figures A1-A6. Additionally, the Table A1 will summarize all the mean values of the standard deviation for all the experiments to give a general overview of the impact of assimilation.

**Code and data availability statement**

The model simulation data and the WRFDA code version 4.4.1, with the gradient operator codes, is available for download. It is stored on Zenodo, a general-purpose open repository developed under the European Open-Access Infrastructure for Research in Europe (OpenAIRE) program and operated by the European Organization for Nuclear Research (CERN). The access link is https://zenodo.org/doi/10.5281/zenodo.13734634.

**Author contributions**

The original draft of the study was written by RT, who also conducted the formal analysis and experiments. FZ and RT collaborated to modify the WRFDA code and develop the gradient operator. JW and GD supervised the project, acquired funding, and reviewed and edited the paper.

**Competing interests**

The contact author has declared that none of the authors has any competing interests.

**Acknowledgements**

The research project is funded by the German Research Foundation (DFG; grant no. 68510200) and is titled "Exploitation of GNSS tropospheric gradients for severe weather Monitoring And Prediction (EGMAP)." The ECMWF conventional datasets for the DA study in this research were provided by Thomas Schwitalla from our collaborative institution, the Institute of Physics and Meteorology, University of Hohenheim, Stuttgart. The GNSS data are provided by the Geodetic Observatory Pecny (GOP) (http://www.pecny.cz (accessed on 25 February 2023)). We thank Michal Kačmařík for preparing the GNSS data set in a user-friendly format.

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

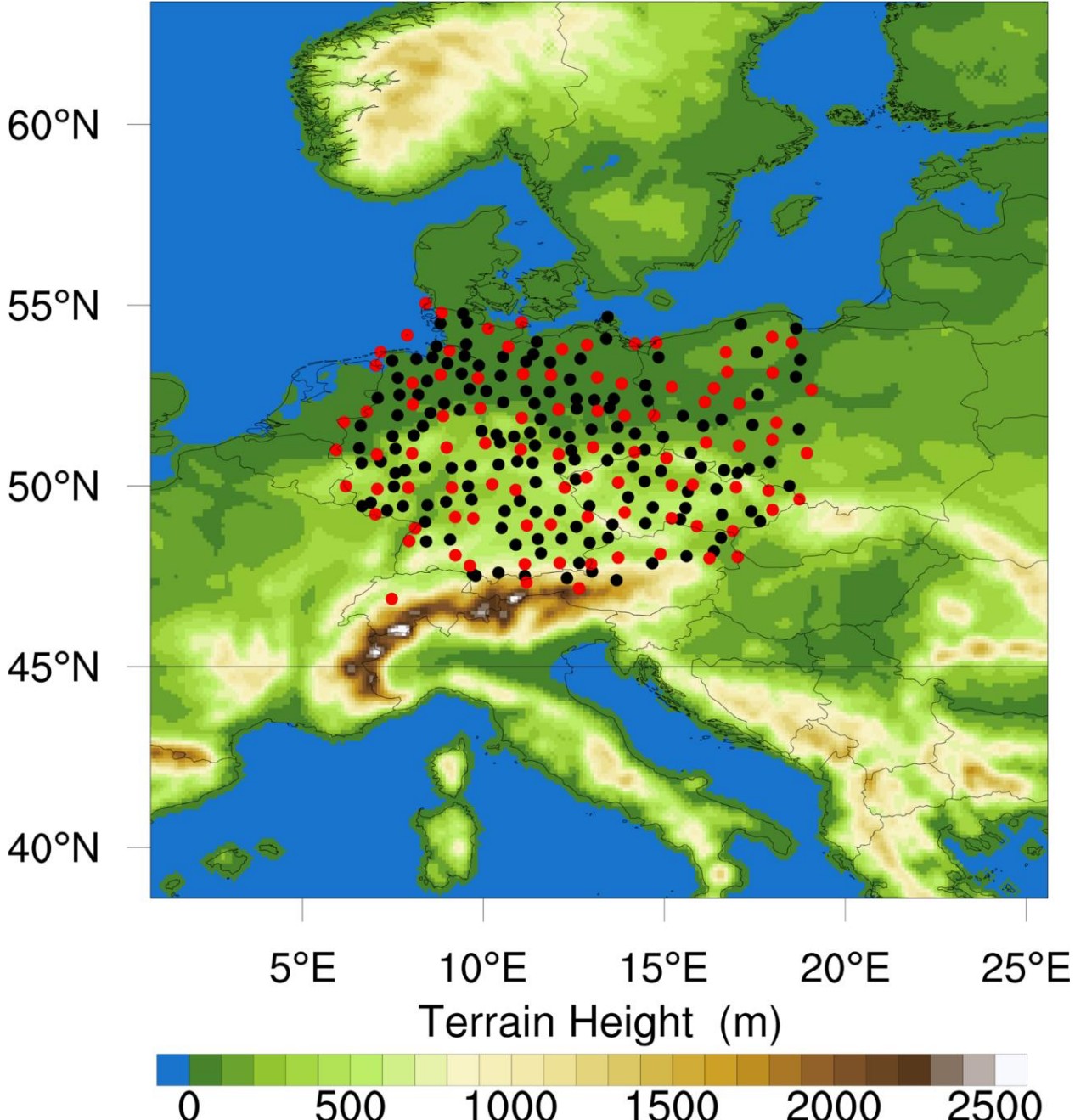

**Figure 1.** The WRF model domain with terrain height representation. The GNSS stations in the assimilation study are depicted in red to signify the sparse network with a 1-degree density, while the combination of black and red indicates the dense network with a 0.5-degree density.

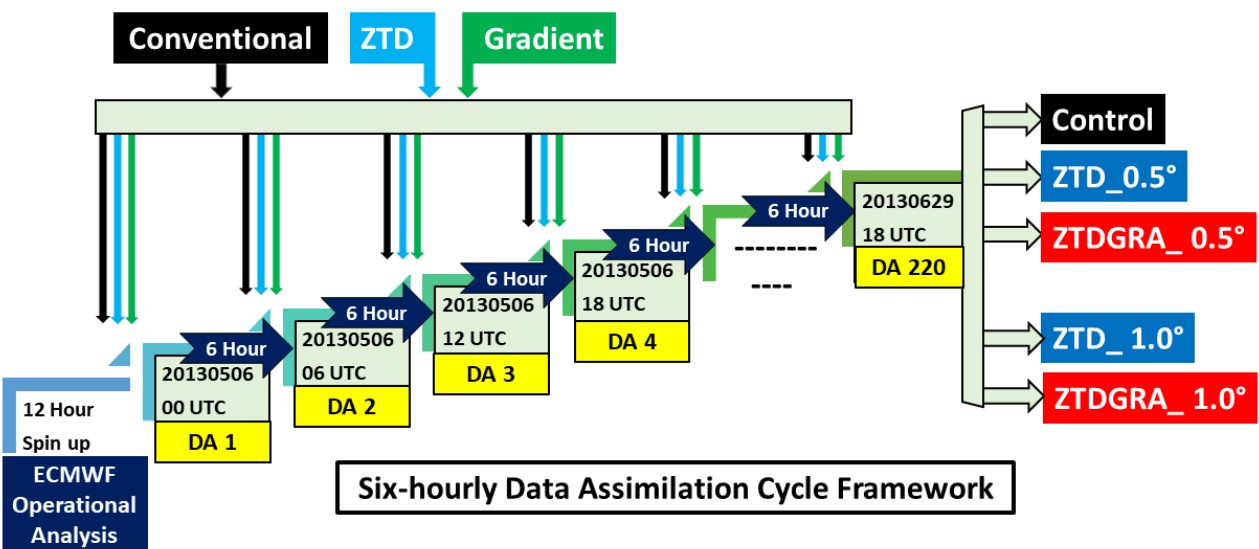

**Figure 2.** Schematic of the 3D-Var Six-hourly DA cycle initialized from the ECMWF operational analysis. Five experiments with different setups are performed in two sets. The first set comprises a control run assimilating conventional data, a ZTD_0.5° run assimilating ZTDs on top of the control run, and a ZTDGRA_0.5° run assimilating ZTD and TGs on top of the control run. These experiments are conducted with the observations from the (dense) 0.5-degree station network. The second set runs are ZTD_1_0° and ZTDGRA_1_0° with the assimilation of observations from the (sparse) 1-degree station network.

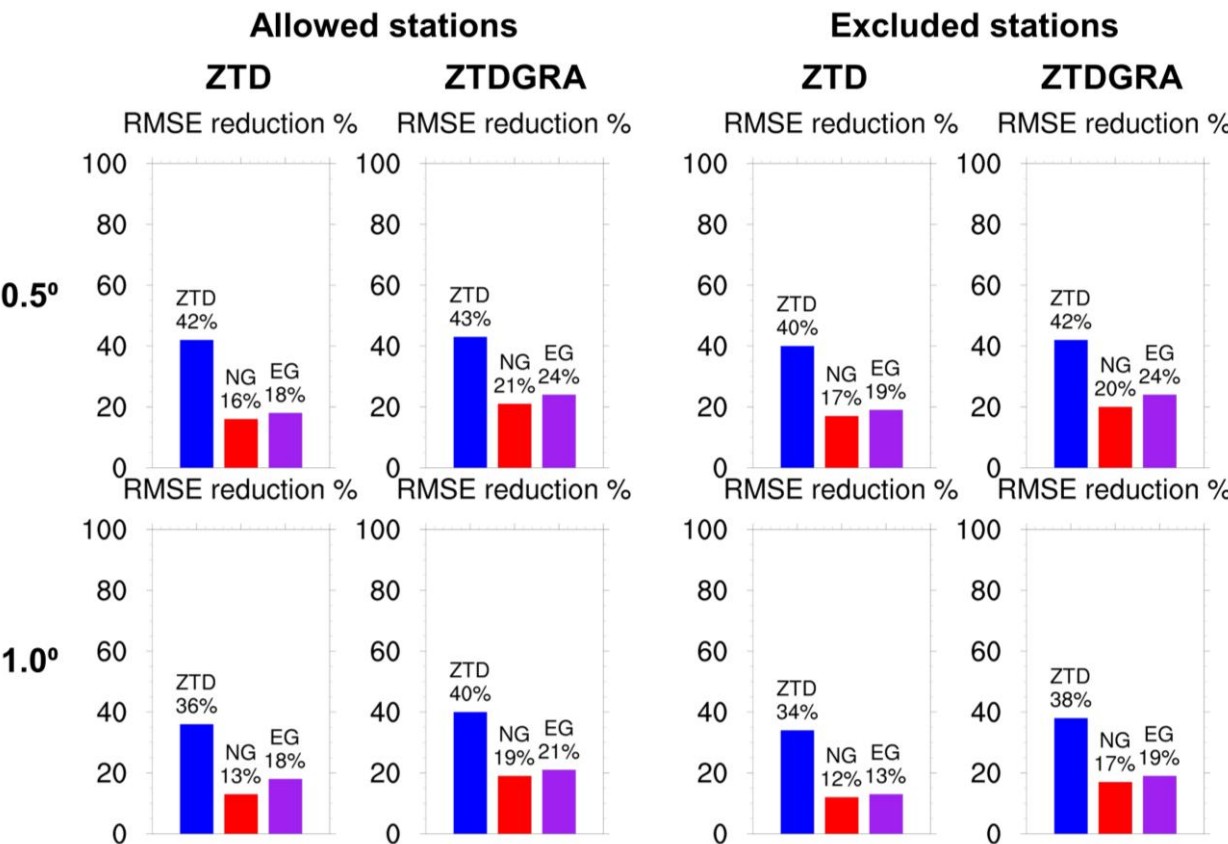

**Figure 3.** RMSE comparison w.r.t stations: assimilated or "allowed" and independent or "excluded." The plot shows the RMSE reduction w.r.t the control run in percentage. The gradient components are termed NG for the North Gradient and EG for the East Gradient. Please refer to the appendices for a detailed plot.

| Allowed stations | | | | | | |
|---|---|---|---|---|---|---|
| **Exp.** | **Standard deviation in mm (0.5°)** | | | **Standard deviation in mm (1.0°)** | | |
| | **ZTD** | **N-Gradient** | **E-Gradient** | **ZTD** | **N-Gradient** | **E-Gradient** |
| **Control run** | 14.4 | 0.62 | 0.66 | 14.4 | 0.63 | 0.66 |
| **ZTD run** | 8.3 | 0.52 | 0.54 | 9.2 | 0.55 | 0.58 |
| **ZTDGRA run** | 8.2 | 0.49 | 0.5 | 8.7 | 0.51 | 0.52 |
| Excluded stations | | | | | | |
| **Exp.** | **Standard deviation in mm (0.5°)** | | | **Standard deviation in mm (1.0°)** | | |
| | **ZTD** | **N-Gradient** | **E-Gradient** | **ZTD** | **N-Gradient** | **E-Gradient** |
| **Control run** | 13.7 | 0.59 | 0.63 | 13.7 | 0.59 | 0.63 |
| **ZTD run** | 8.2 | 0.49 | 0.51 | 9 | 0.52 | 0.55 |
| **ZTDGRA run** | 8 | 0.47 | 0.48 | 8.5 | 0.49 | 0.51 |

**Table 1.** Mean standard deviation derived out of station specific standard deviation. The allowed stations and the excluded stations are compared. The grey background signifies that the values are close for ZTD dense network and ZTDGRA sparse network.

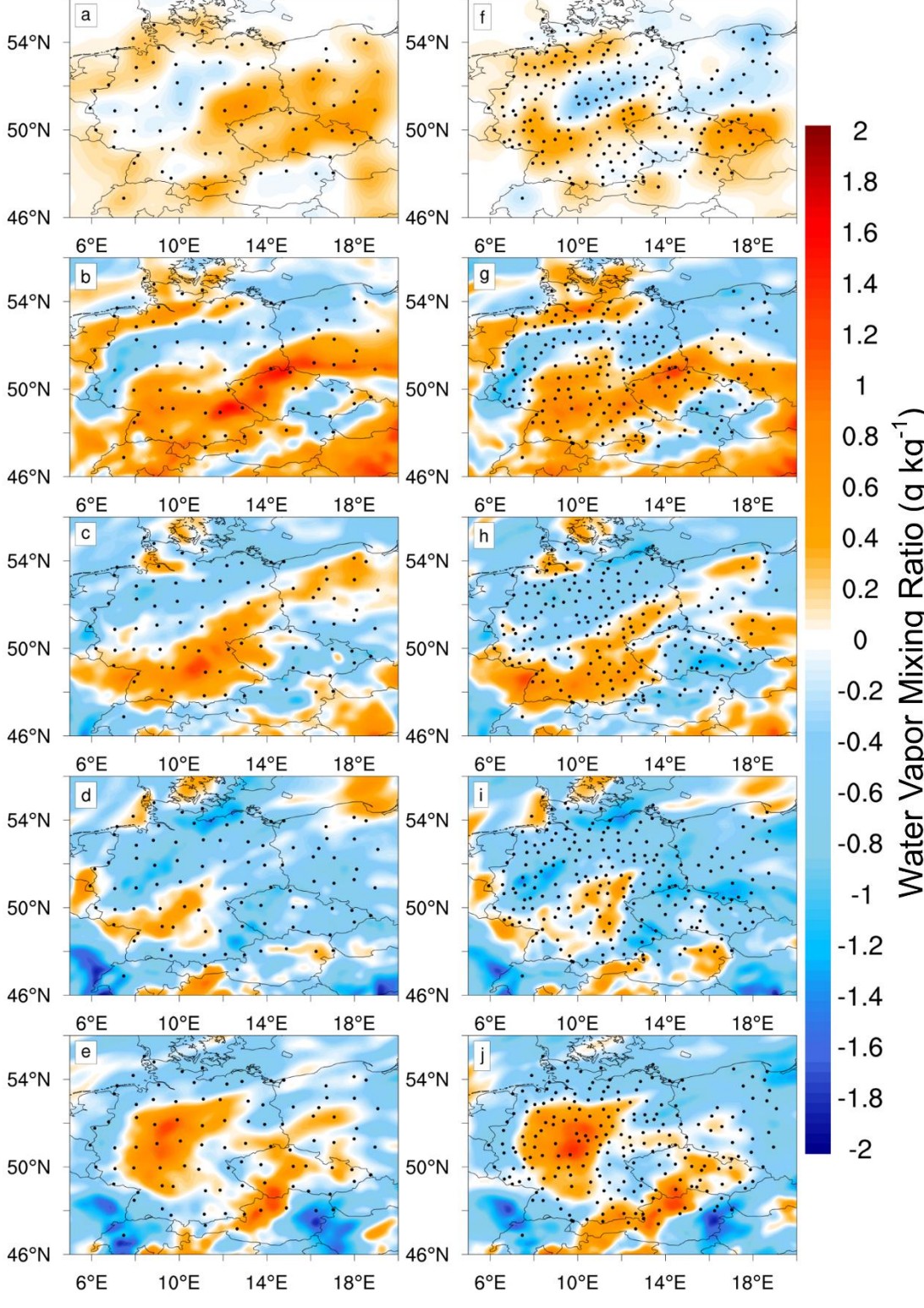

**Figure 4.** Spatial comparison of the evolution of the analysis increments of ZTDGRA_1.0° (left column) and ZTD_0.5° (right column) runs for the first five assimilation cycles. The stations used for the respective assimilation runs are depicted
by black dots.

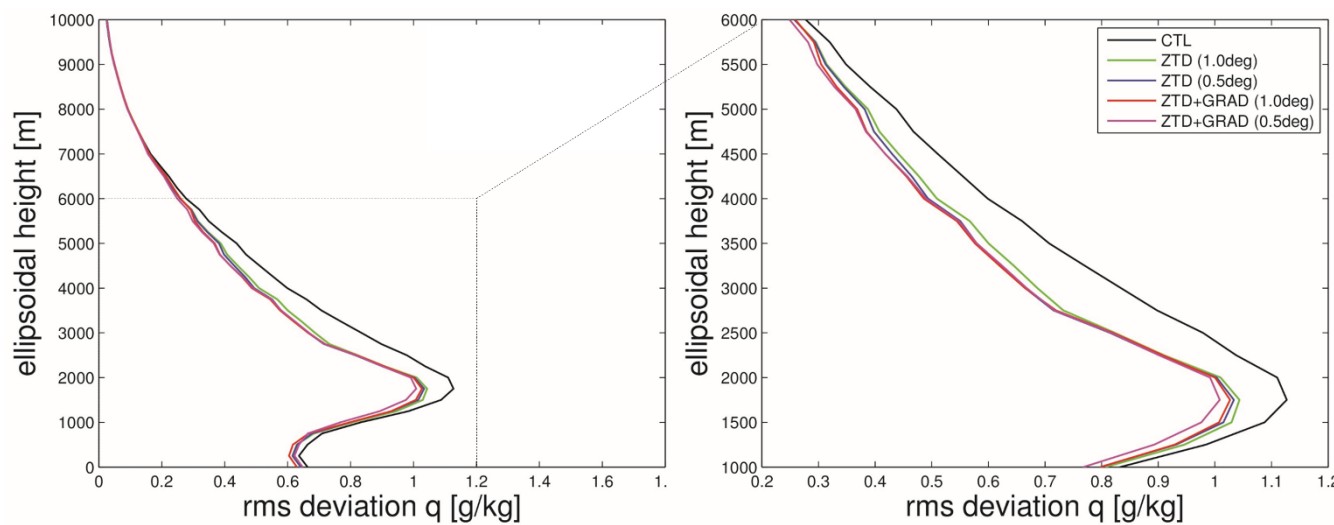

**Figure 5.** The RMSE of specific humidity profiles compared to ERA5 for Control run (black), ZTD_0.5° run (blue), ZTD_1.0° run (green), ZTDGRA_1.0° run (red), and ZTDGRA_0.5° run (purple). Profiles were compared at five selected
stations for 220 DA cycles, totaling 1100 profiles for the average plot.

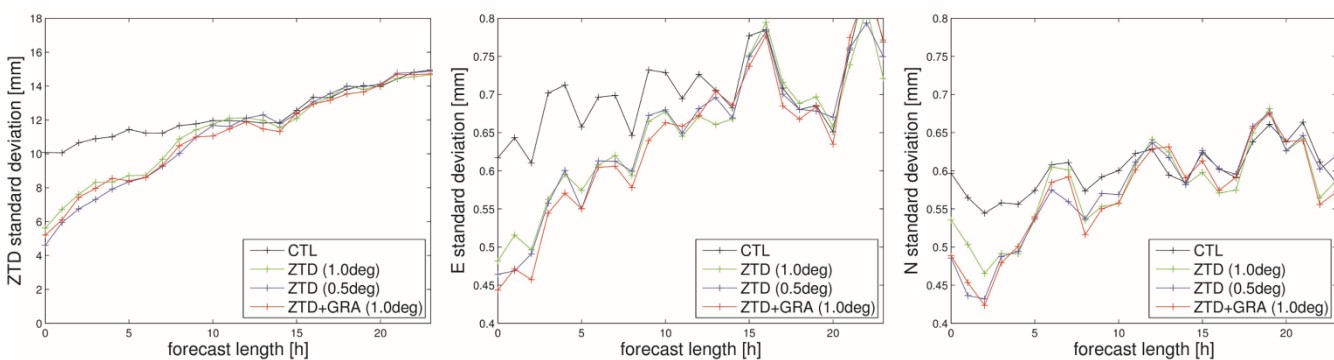

**Figure 6.** Average forecast impact with a 24-hour lead time initiated from 12 analyses over 3 days starting from 6 May 2013 00 UTC. The control run (black), ZTD_1.0° run (green), ZTD_0.5° run (blue), and ZTDGRA_1.0° run (red) forecasts are compared to independent GNSS stations. Subplots- **Left:** ZTD standard deviation; **middle:** East Gradient standard deviation; and **right:** North Gradient standard deviation.

## Appendix A

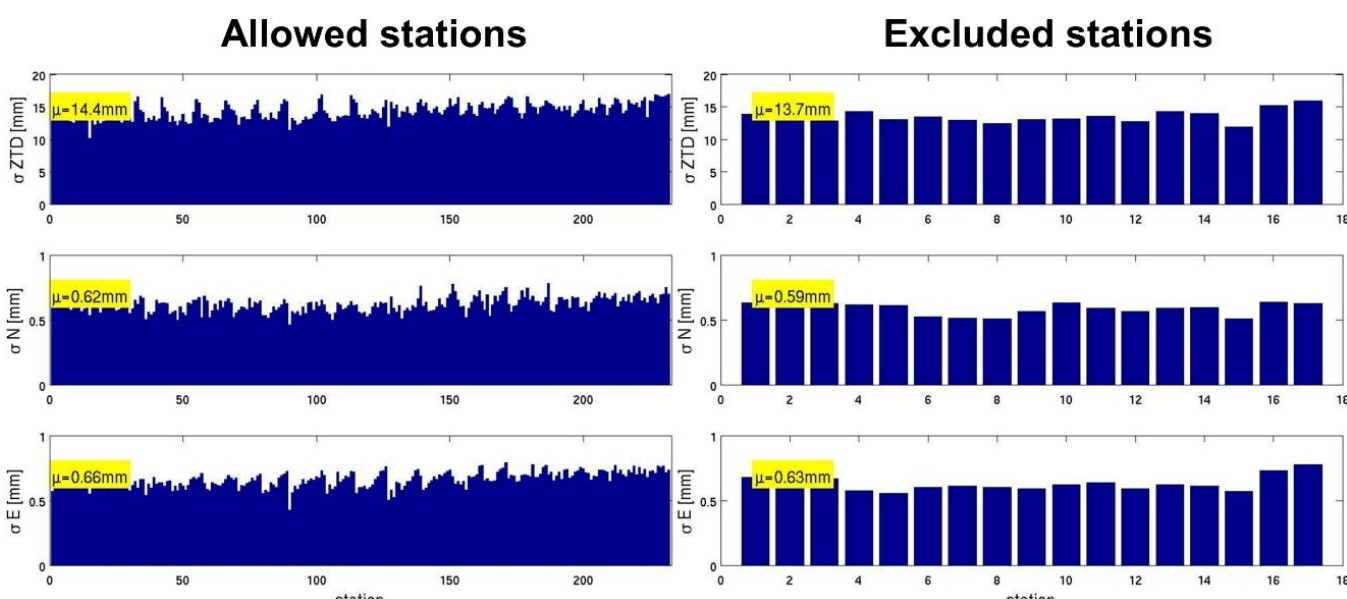

**Figure A1.** Station specific standard deviation of the Control run with dense network configuration. ZTD, North gradient, and East gradient of the allowed and excluded stations are shown on the left column and right column, respectively. The mean values are shown as text with the yellow background.

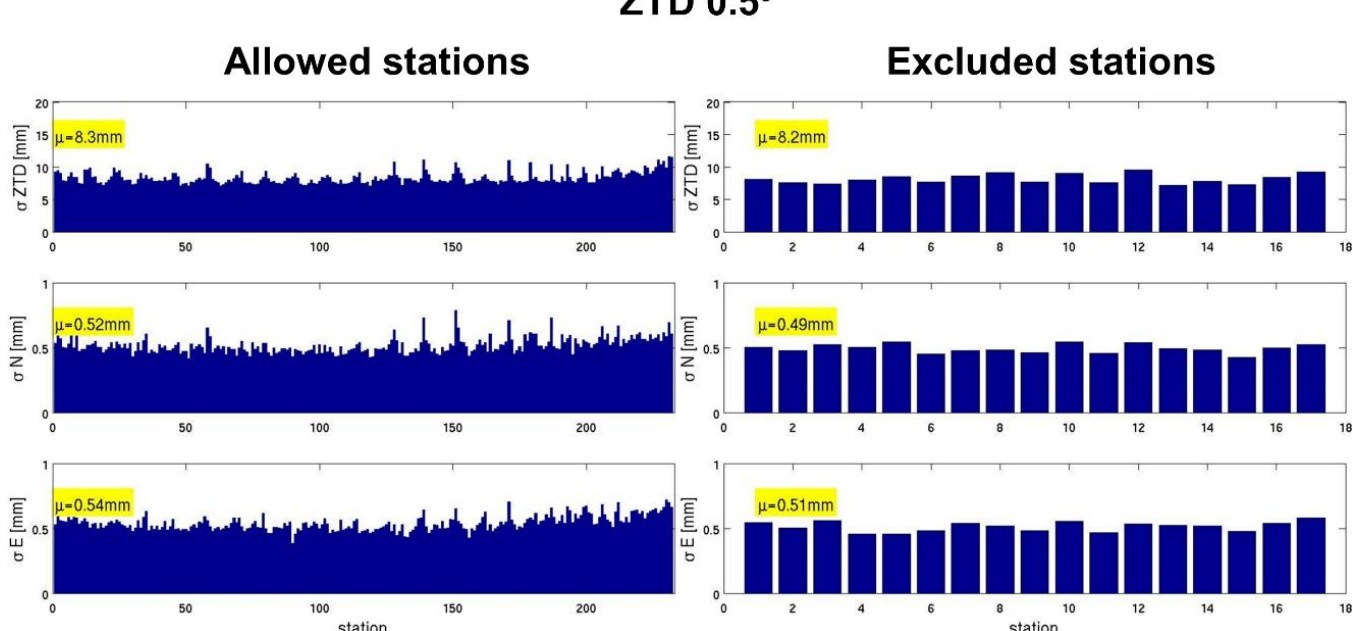

**Figure A2.** Same as Figure A1, but for ZTD run in the dense network configuration.

## ZTDGRA 0.5⁰

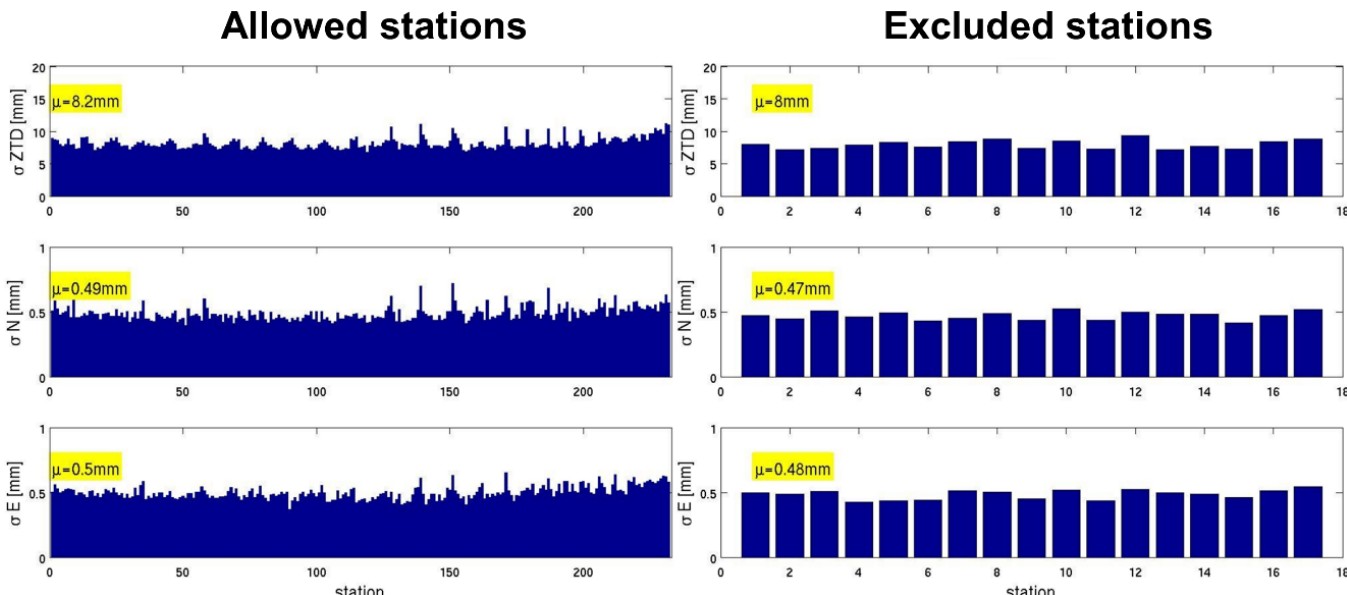

**Figure A3.** Same as Figure A1, but for ZTDGRA run in the dense network configuration.

## CONTROL 1.0⁰

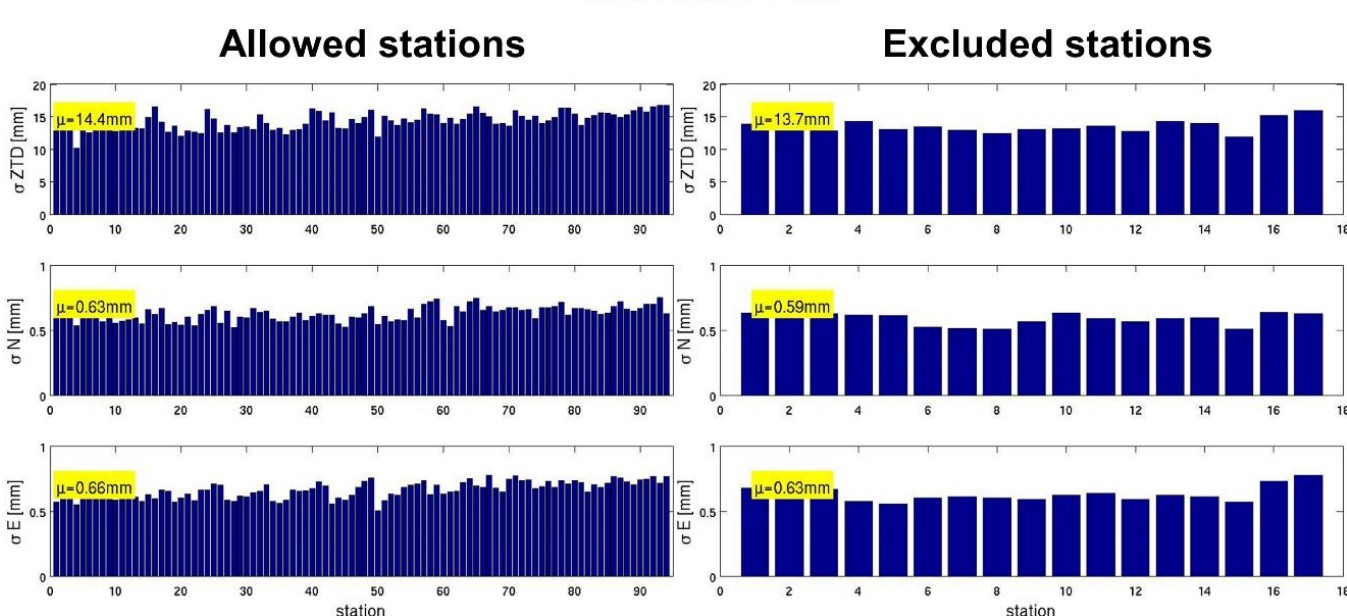

**Figure A4.** Same as Figure A1, but for Control run in the sparse network configuration.

## ZTD 1.0⁰

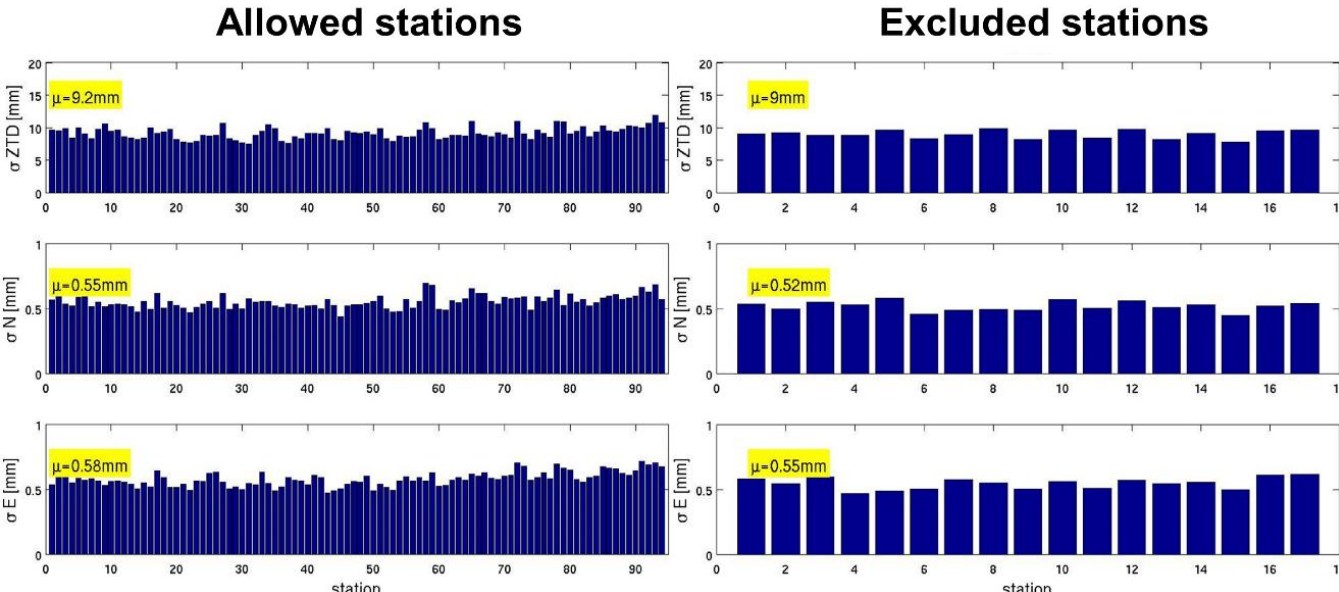

Figure A5. Same as Figure A1, but for ZTD run in the sparse network configuration.

## ZTDGRA 1.0⁰

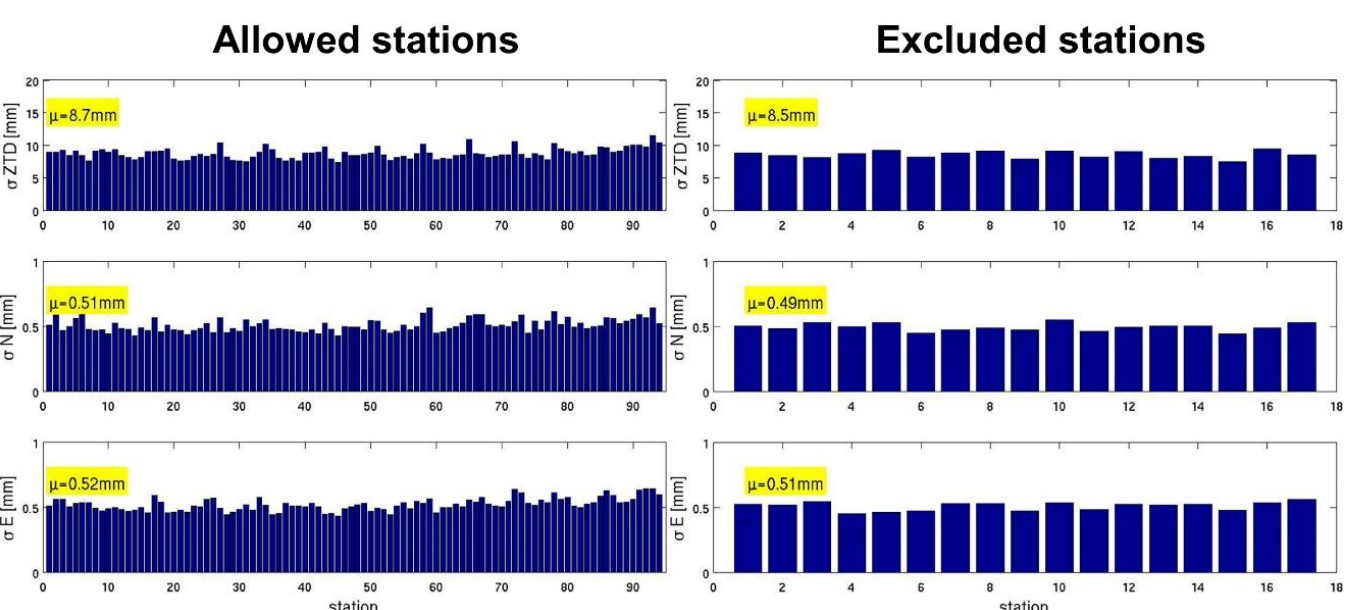

**Figure A6.** Same as Figure A1, but for ZTDGRA run in the sparse network configuration.