# Peer review of "Assimilation of GNSS Zenith Delays and Tropospheric Gradients: A Sensitivity Study utilizing sparse and dense station networks"

_EGUsphere, 2025_

## Author Response (AR1)

Dear Editor,

Thank you very much for considering our manuscript for the review process. We would like to take this opportunity to thank all the reviewers for providing their valuable comments and suggestions to improve the manuscript. We strongly believe that the manuscript has improved from the first version, and we were able to incorporate almost all the comments from the reviewers.

Thank you once again.

Apart from the reviewer comments, we have made a few changes to the manuscript, which we would like to list below. Please refer to **author track-changes file.**

**Line 5:**

The affiliation has been changed to "GFZ Helmholtz Centre for Geosciences"

**Line 90:**

Replaced "Nevada Geodetic Laboratory (NGL)" to "GFZ Helmholtz Centre for Geosciences."

**Line 175:**

The sentence below was removed since TAMDAR observations were not used in this study. We apologize for the mistake in the first version.

"To address the underrepresentation of the radiosonde network during specific periods, such as 06:00 and 18:00 UTC rather than 00:00 and 12:00 UTC, we used a series of Tropospheric Airborne Meteorological Data Reporting (TAMDAR) observations."

**Line 435:**

The Li et al. (2015) reference was replace. The reference was not correct in the previous version.

Li, X., Zus, F., Lu, C., Ning, T., Dick, G., Ge, M., ... & Schuh, H. (2015). Retrieving high-resolution tropospheric gradients from multiconstellation GNSS observations. *Geophysical Research Letters*, *42*(10), 4173-4181. doi: 10.1002/2015GL063856

From the next pages, all the replies to the reviewer comments are attached.

Thank you and best regards,

Rohith Muraleedharan Thundathil

**Note: All Line numbers from here on refer to the No Markup version of the revised Manuscript.**

**Reviewer #1**
Interesting and well-written report, however further insight could have been good, for example using the Desroziers method to quantify the relative impact of the different observations used in the experiments.

Thank you for your kind feedback and comments. In response, we have now included a quantification of the relative impact using the Desroziers method. We have computed Desroziers statistics for 220 data assimilation cycles using SYNOP data, ZTDs, and Tropospheric Gradients (for both the North and East gradients).
Please refer **Lines – 250 to 267**
"To quantify the relative impact of GNSS observations compared to other point observations in the study, specifically the SYNOP station data, we utilized the Desroziers method. Desroziers method is an effective diagnostic tool used to evaluate the impact of various observations. By analyzing the Innovation (Observation minus Background, OMB) and Residual (Observation minus Analysis, OMA) statistics, we can estimate the covariances of observation and background errors. This analysis helps us determine the relative influence of different types of observations on the overall analysis.
The relative impact of an observation is determined by the ratio of the estimated observation error covariance R to the estimated background error covariance B. The respective error covariances are calculated as below:

$$R = E[(y - Hx^a)(y - Hx^a)^T]$$

$$B = E[(y - Hx^b)(y - Hx^a)^T]$$

Here $(y - Hx^b)$ is the innovation and $(y - Hx^a)$ is the residual where $x^a$, $x^b$, and $y$ are the model state vectors for analysis, background and observations, respectively.
The higher the value of the ratio $B/R$, the higher the impact of the observation. The observations likely to enhance the model or lead to effective assimilation fall within the range of 0.5 to 3. A value below 0.5 suggests that the observation has large error, making it unreliable for assimilation. Conversely, values above 3 indicate that the observation forces the background towards the observation, which may result from a small observation error or a bias in the observation. After analyzing 220 DA cycles, the average $B/R$ ratios were as follows: SYNOP at 1.4, ZTD at 2.8, NG at 1.8, and EG at 1.5. These values indicate that the impacts of the observations are well within the acceptable range. Additionally, the TGs, have an impact in the assimilation system. The North and East gradient values indicate that the assimilation was effective. The ZTD observation has higher values, which might indicate that the observation error assigned to ZTDs could be higher than the current observation error value of 8 mm."

row 72 is vague as it proposes that ZTD is the only source of moisture data used operational and should be clarified!
Thank you for the comment. The sentence was incomplete which resulted in a different meaning. The sentence in now corrected as below. Please refer to **Line 72**.
"ZTDs are the only GNSS-derived moisture data used operationally; however, they provide limited atmospheric information."

**Reviewer #2**

This is an excellent manuscript, almost ready to publish.

It addresses a topic that is currently at the forefront of GNSS meteorology: The use of ZTD delay gradients estimated from GNSS data in numerical weather prediction models via data assimilation, on top of assimilation of ZTD.

By consideration of both a dense and a sparse GNSS network dataset, it is demonstrated that assimilation of ZTD gradients (which are cheap to obtain) improves the resulting analyses in a way that would otherwise require a denser network (which is in most cases expensive to obtain) if only ZTD is assimilated.

A few things should be improved:

Specify in more detail the type and amount of "conventional data" that was assimilated.

Thank you for the comment. We have now clarified the type and amount of conventional data used in the experiments.
Please refer to **Lines 170 to 174**.
"To improve the analysis, we assimilated a set of conventional observations in addition to the GNSS observations. The conventional observations included a network of SYNOP stations across Europe. Radiosonde measurements offered a detailed view of the atmospheric thermodynamic structure at launch points. In order to maintain simplicity within the DA system, we limited conventional datasets to SYNOP surface observations and radiosondes. The number of observations ranges from 1029 to 1225 for SYNOP stations and 4 to 35 for radiosondes."

In line 73. Consider changing to: "..only source of GNSS moisture data..", unless the conventional data did not include any humidity data.

Thank you for the comment. The sentence has been corrected. Please refer to the second comment from reviewer #1.

Figure 2 is confusing. As I understand the text DA is done every 6 h. I take it is done simultaneously for the different types of observations (depending on the experiment), using a 6 h old forecast from the previous run with that experiment as first guess.
But why are then the conventional observations represented by tilting arrows, and the GNSS data by vertical. Somehow horisontal is time in the figure, but the arrows are separate in time then.
I would expect there to be a "DA 2" in the box named 20130506 06 UTC, a "DA 3" in the box named 20130506 12 UTC, and so forth.

Thank you for the comment. We have now changed the Figure 2 as per the reviewer's comments.

[Figure]

**Figure 2.** Schematic of the 3D-Var Six-hourly DA cycle initialized from the ECMWF operational analysis. Five experiments with different setups are performed in two sets. The first set comprises a control run assimilating conventional data, a ZTD_0.5° run assimilating ZTDs on top of the control run, and a ZTDGRA_0.5° run assimilating ZTD and TGs on top of the control run. These experiments are conducted with the observations from the (dense) 0.5-degree station network. The second set runs are ZTD_1_0° and ZTDGRA_1_0° with the assimilation of observations from the (sparse) 1-degree station network.

**Reviewer #3**

Review comments to the paper "Assimilation of GNSS Zenith Delays and Tropospheric Gradients: A Sensitivity Study utilizing sparse and dense station networks", manuscript ID egusphere-2025-19.

General comments:

The topic of the paper is really interesting and could potentially be very valuable for areas with sparse networks of GNSS receivers. I write "potentially" since from what I read from the paper it is not possible to determine very much from the results. More specific comments will follow but as the paper is written it is not suitable for publication. In fact it is rather poorly written with very little explanations of figures and results which makes it impossible to understand what is plotted and how it should be interpreted. The recommendation is therefore major revisions with a much better description of what is verified and why. I would as well like to see results for forecasts longer than 5 hours.

Specific comments:

1. A minor comment from the introduction: "ZTDs are the only source of moisture data used operationally" – This is not true. Moisture is obtained from radiosondes as well as from several satellite observations. Nowadays, many operational NWP models also include relative humidity at 2 meter height from synop stations.

Thank you for the comment. The sentence has been corrected. Please refer to the second comment from reviewer #1.

2. Line 123: Does entire Figure 1 show the model domain? If so (1), there is a lot of area that is not covered by GNSS observations. If so (2), it does not match the domain shown in Figure 4.

Yes the entire Figure 1 shows the model domain. We have now changed the text in the manuscript to clarify the Figures 1 and 4.

Please refer to **Lines 123-127.**
"In this sensitivity study, we are using the GNSS observations from the Benchmark data set which was collected within the European COST Action ES1206 GNSS4SWEC (Advanced GNSS tropospheric products for monitoring severe weather and climate; Douša, Jan, et al., 2016). The GNSS stations in central Europe covering Germany, the Czech Republic, and part of Poland and Austria provided the data during this campaign."

Please refer to **Lines 298-299.**
"Figure 4 is a zoomed-in map that covers only the countries where the GNSS stations were located."

3. Line153: The CV5 option does not mean anything unless you are familiar with WRF. Explain further or remove.

Yes we agree with the reviewer. The CV5 option has been elaborated for the readers.

Please refer to **Lines 186-192.**
"The **B** matrix for the regional simulations was derived from the forecast statistics by analyzing differences in 24-hour and 12-hour predictions over a month using data from May 2013. We chose the CV5 option for independent control of moisture levels, as it minimizes interference from other control variables. CV5 refers to a version of the background error covariance matrix used in the WRF model. It incorporates five control variables: stream function ($\Psi$), unbalanced velocity potential ($\chi_u$), unbalanced temperature ($T_u$), pseudo-relative humidity ($RH_s$), and unbalanced surface pressure ($P_{s,u}$). Pseudo-relative humidity is the ratio of $Q$ to $Q_{b,s}$, where $Q_{b,s}$ represents the saturated specific humidity of the background field."

4. Line 154: A 6 hour assimilation cycle is not a rapid update cycle. In fact it is the opposite. Most NWP models nowadays run with 3 hour cycling or a rapid update cycle of 1 hour.

Yes we agree with the reviewer. Rapid update cycle has been removed and replaced with "Six-hourly DA cycle".

5. Lines 155-165: This is a bit hard to follow since the authors have named the experiments after the included observations. The observations are described in the following section. I would suggest to describe the observations first, i.e. switch sections so that 3.2 becomes 3.1 and vice versa.

Thank you for the comment. The flow of the manuscript section 3 has now been restructured.

6. Line 165, Figure 2. A spin up of 12 hours is mentioned in the figure caption. Why is this spin up run and is 12 hours enough? It should be explained in the text and not in the figure caption.

Yes we agree with the reviewer. Now there is an explanation in the manuscript and not in the figure caption.
Please refer to **Lines 193 to 196.**
"Spin-up is essential for the model to stabilize with the initial and boundary conditions, enabling it to respond accurately to any desired inputs. Only after a sufficient spin-up period can the model forecasts be considered reliable for further analysis through data assimilation (DA). For our study, we adopted a 12-hour spin-up before the assimilation (Lauer et al., 2023)."

**Reference included:**
Lauer, A., Devaney, J., Kieu, C., Kravitz, B., O'Brien, T. A., Robeson, S. M., ... & Vu, T. A. (2023). A convection-permitting dynamically downscaled dataset over the Midwestern United States. Geoscience Data Journal, 10(4), 429-446.

7. Line 159: Why only run 5 hour forecasts? In an impact experiment you would normally want to run at the very least 12 hour but up to 24 is recommended in order to see how persistent the impact from the

observations is. If there is a big impact on the analysis and very short forecasts that quickly disappears it can be an indication that your observation error is wrong and you give too much weight to the observations.

Thank you for the comment. As per the reviewers' comment we have now included a 24 hour forecast from the analysis for 12 DA cycles. We have included a new figure which depicts the observation impact forecast for 24 hours. The impact of the assimilation holds for not more than 12 hours which is quite reasonable for moisture data assimilation. We have now introduced a section 4.3 in the results as below:

"**4.3 Forecast impact**
To understand how long the effects of GNSS observations assimilation persist within the model, we conducted simulations of 24-hour forecasts based on a three-day analysis. Each day included four assimilation cycles, resulting in a total of 12 forecasts, each covering 24 hours. The forecast is better validated with independent observations that are not assimilated into the model. With the 18 excluded GNSS stations, we can directly compare the model forecast with observations from the GNSS stations. Figure 6 compares the 12 forecast average with the GNSS ZTDs and TGs, including the North and East Gradients, to compute the standard deviation. We analyzed three impact experiments: ZTD_1.0°, ZTD_0.5°, and ZTDGRA_1.0°, in addition to the control run. As anticipated, the effects of the three impact experiments gradually diminish and converge with the control run. If we define the endpoint of the impact as the moment when the standard deviation of the impact experiment aligns with that of the control run, then the duration of the impact is 12 hours. The effects of the assimilation last for no more than 12 hours, which is quite reasonable for moisture data assimilation. Additionally, it is important to note that incorporating TGs along with ZTDs enhances the forecast. Furthermore, the forecast impact of ZTDGRA_1.0° is comparable to that of ZTD_0.5°."

[Figure]

**Figure 6.** Average forecast impact with a 24-hour lead time initiated from 12 analyses over 3 days starting from 6 May 2013 00 UTC. The control run (black), ZTD_1.0° run (green), ZTD_0.5° run (blue), and ZTDGRA_1.0° run (red) forecasts are compared to independent GNSS stations. Subplots- **Left:** ZTD standard deviation; **middle:** East Gradient standard deviation; and **right:** North Gradient standard deviation.

8. Section 3.2: Why was the resolutions 0.5 degrees for the dense network and 1.0 degrees for the sparse network selected?

Thank you for the comment. We have now elaborated with additional sentences.
Please refer to **Lines 148 to 156.**
"To ensure a homogeneous set of observations across the domain, we excluded collocated and clustered stations and specifically chose GNSS stations with data availability exceeding 75%. In addition to complying with our WRF model domain, we carried out a simple thinning of observations ('homogenization' of station distribution). **The thinning method was conducted in two steps. First, a 0.5-degree mesh was constructed. Then, GNSS stations were selected based on their proximity to the mesh-grid point. Finally,** we obtained a station network with a resolution of about 0.5 degrees. After these steps, we were left with around 250 GNSS stations over the Benchmark domain. For the sensitivity experiment, we created another thinned station network with a resolution of about 1 degree that contained around 110 stations (see Fig. 1). **The same thinning procedure was used again.**"

9. Line 178: "a station specific bias correction" – Is this a variational bias correction or a fix one? In case of variational, what are the predictors and if fixed, how was it derived?

Thank you for the comment. We have now added a sentence to clarify the point.
Please refer to **Lines 161 to 162.**
"We utilized analyses from our control experiment to implement a "fixed" bias correction, addressing potential biases in the GNSS dataset (Thundathil et al., 2024)."

10. Lines 179-180: The observation errors, how were they selected or derived?

Thank you for the comment. We have now added an explanation as follows:
Please refer to **Lines 164-170.**
"The standard observation error for ZTDs in operational forecasting typically ranges from 5 to 15 mm. Similar to our previous study, Thundathil et al. (2024), the same observation errors were adopted: 8 mm for the ZTDs and 0.65 mm for the TGs. Given the high quality of the observations from the Benchmark campaign dataset, we have maintained these same error values in this study. The North and East gradient observation errors were calculated based on an analysis of the observation-minus-background (OB) statistics from the control run. OB statistics encompass both observation and model errors. An observation error of 0.65 mm was conservative since we did not want to force the model too much to the observations."

11. Line 180: "we set up a thorough network of surface reports" – What does this mean? That you install your own observations?

Thank you for the comment. We understand that the sentence did not convey the message due to its inaccurate construction. The sentence is now clarified as below. Please refer to **Lines 170 to 171**.
"To improve the analysis, we assimilated a set of conventional observations in addition to the GNSS observations. The conventional observations included a network of SYNOP stations across Europe."

12. Entire section 4: The authors compare the RMSE and the improvement of RMSE. The RMSE of what? Is is the comparison of the analysis and observations or forecast and observations? If the latter, what forecast lengths?

Thank you for the comment. We have now included an explanation how the RMSEs were calculated. Please refer to **Lines 216 to 220.**

"The model simulation for two months in each experiment comprises six-hourly analyses and a five-hour forecast in between two DA cycles. Hence, the model simulations consist of hourly model outputs of analyses and forecasts. This hourly model output is compared to the corresponding GNSS station data for each experiment to calculate the root mean square error (RMSE). The model ZTDs and TGs at the locations of each specific station are computed for the RMSE. We term this the "station-specific RMSE.""

13. Lines 193-204 and Figure 3: The authors state that they "clearly demonstrate" a positive impact. I don't really see how. First, Figure 3 shows RMSE change (again of what) in percent while the text describes the same change in mm. Please be consistent. Secondly, the figure caption need to describe the figure better, e.g. what is NG and EG (one could guess but still)?

Thank you for the comment. The word "clearly" has now been removed. The results are now explained more clearly.
Please refer to **Lines 222 to 227.**

"Figure 3 is simplified in a percentage analysis of the station-specific RMSE plot (please refer to Figures A1-A6 in the appendices) for a more straightforward interpretation. Here, the control experiment was kept as the base experiment, and the ZTDs and ZTDs plus TGs assimilation experiments in dense and sparse configurations were compared. A reduction in the RMSEs indicates improvement in the assimilation experiment. The reduction in RMSE is represented as a percentage increase, which means the higher the reduction, the higher the percentage. Table 1 lists the average of the station-specific RMSEs of all the DA experiments for the two months."

Also the Table A1 in the Appendices section has been moved to the main section for reference to the mm values of the mean station-specific RMSEs.

14. Line 206: "we extend the analysis to include 18 independent GNSS stations" – Does this mean the these are in addition to the other stations or instead of the other stations? And why only 18? There are 430 available stations, you select 250 which means you should have an additional 180 stations to use for verification.

Thank you for the comment. We have now briefly answered the idea behind choosing the 18 stations. Please refer to **Lines 157 to 161.**

"In line with the approach of Thundathil et al. (2024), we intentionally excluded 18 stations from our dataset for validation purposes. These excluded stations were chosen strategically to maintain a balanced spatial distribution, aligning with the locations of the German Weather Service (DWD) radar stations. The remaining stations included in the model are referred to as "allowed" stations. This method enabled us to analyze improvements with respect to independent observations."

15. Line 214: "assimilation of ZTD and TGs significantly enhances that accuracy" – Have you tested the significance or is it just a feeling? If you write this it needs to be validated.

We agree with the reviewer. More evaluation is needed for this statement. We would like to change the statement to the following. Please refer to **Lines 248 to 249.**
"The two-month-long statistical evaluation confirms that the combined assimilation of ZTDs and TGs improves the humidity field."

16. Line 234: Again, significant, what does it mean?

We have now removed the word significant. Please see the previous comment.

17. Figure 4: The unit should be at the large color bar. The panels are labeled a-j but these are never referred to.

Thank you for the comment. This has now been rectified. Please refer to **Lines 300 to 301.**

"DA cycle 1 refers to Figures 4a and 4f, and DA cycles two, three, four, and five refer to Figures 4b and 4g, Figures 4c and 4h, Figures 4d and 4i, and Figures 4e and 4j, respectively."

18. Line 246: Perhaps it could be of interest to separate the very lowest model levels and the slightly higher ones, i.e. separate the boundary layer and the free atmosphere?

Yes, we agree with the reviewer that it could be of interest. We would like to publish further analysis on the vertical distribution of the humidity and the influence of tropospheric gradients in a future manuscript.

19. Lines 251-252: SSIM index parameter – Please explain shortly what this is and how it is calculated. If the reader is not familiar with SSIM the number 0.98 does not mean much without reference.

Yes, thank you for the comment. We have now described the SSIM index and how it is computed. Please refer to **Lines 307 to 315.**
"The Structural Similarity Index (SSIM) is a metric used to quantify the similarity between two images (Wang et al., 2004). Here is a short explanation of the computation of SSIM in our study:

$$SSIM(A,B) = \frac{(2\mu_A\mu_B + c_1)(2\sigma_{AB} + c_2)}{(\mu_A^2 + \mu_B^2 + c_1)(\sigma_A^2 + \sigma_B^2 + c_2)}$$

Here A and B represent the images in the left column and the right column respectively. WVMR is the moisture variable presented here in the images with a span of 101 colors. $\mu_A$ and $\mu_B$ are the mean, $\sigma_A^2$ and $\sigma_B^2$ are the variance, and $\sigma_{AB}$ is the covariance. The variables $c_1$ and $c_2$ are computed based on the color span in the images.

$$c_1 = (k_1L)^2$$
$$c_2 = (k_2L)^2$$

Here $k_1$ and $k_2$ are 0.01 and 0.03 by default. $L$ here is the total number of colors in the color bar minus one. Hence the values of $c_1$ and $c_2$ comes to 1 and 3."

20. Figure 5: Again, what is shown? RMSE of what?

Thank you for the comment. The explanation in provided in the manuscript as below. Please refer to **Lines 317 to 320.**
"In order to analyze the humidity profile correction in the assimilation experiment, we computed the RMSE of specific humidity profiles from model simulations with respect to ERA5 at five locations spread equidistantly across the domain (for details, see Thundathil et al., 2024)."

Please refer to the updated Figure caption:
"**Figure 5.** The RMSE of specific humidity profiles compared to ERA5 for Control run (black), ZTD_0.5° run (blue), ZTD_1.0° run (green), ZTDGRA_1.0° run (red), and ZTDGRA_0.5° run (purple). Profiles were compared at five selected stations for 220 DA cycles, totaling 1100 profiles for the average plot."

**Reviewer #4**

General Remarks

The manuscript provides a relevant overview of the state of the art regarding the assimilation of ZTD and tropospheric gradients, although the operational use of ZTD in major NWP centers could have been detailed further. The evaluation of the complementarity of gradients is an interesting topic, and the idea of two observation densities is insightful. The additional impact of gradients seems very positive and advocates for operational use, but the protocol implemented is not sufficiently detailed, and the results are sometimes lacking in clarity. The following review raises several questions, the answers to which could be usefully added to the manuscript.

Questions about the Dataset Used

- What conventional data were used?

Thank you for the comment. We have now written it more clearly in the manuscript. Please refer to the first comment of reviewer #2.

- Was it possible to use satellite data in the assimilation cycle? If so, why did you choose not to use them?

Thank you for the comment. Our goal in this study is to understand the impact of GNSS observations, i.e., the added value of assimilation TGs on top of ZTDs. However, assimilation of further data, such as satellite and radar, would be of interest in a future study.

- How was the geographical thinning performed to reduce the resolution of observation sets? Was it based solely on interdistance or also on quality criteria?

Thank you for the comment.
Please refer to the comment 8 of the reviewer #3 for a detailed explanation.

- Did you conduct prior monitoring of ZTD and gradient data? If so, what were the rejection thresholds in RMS and bias? Otherwise, what type of quality control is performed in the assimilation cycle?

Thank you for the comment. We have provided an explanation in the manuscript for the quality control of the observations prior to assimilation as follows. Please refer to **Lines 148 to 150.**
"To ensure a homogeneous set of observations across the domain, we excluded collocated and clustered stations and specifically chose GNSS stations with data availability exceeding 75%."

- Why did you choose such a limited sample of excluded stations?

Thank you for the comment. Please refer to the comment 14 of reviewer #3 for the explanation.

Questions and Comments on the Impact of the Data

- Why did you limit the forecast range to that of the assimilation cycle? This seems insufficient for drawing solid conclusions, knowing that a positive short-range impact calculated in the space of added observations can be misleading and unsustainable. At a minimum, forecasts with a 24-hour range should be performed to draw definitive conclusions.

Thank you for the comment. Please refer to comment 7 of reviewer #3 for a new figure depicting 24 hour forecast impact.

- The reference for RMSE calculations is unclear. It should be specified in every occurrence within the text and figures what is actually being calculated.

Thank you for the comment.
Please refer to comments 12 and 20 of reviewer #3 for detailed explanation of RMSE calculation in the two scenarios of the manuscript.

- Unless I missed it, why did you not use radiosonde data to evaluate the forecast impact in the various configurations?

Thank you for the comment. However we would refrain from deviating the main idea of the manuscript which is the sensitivity analysis of the TGs. For a detailed comparison to radiosondes we have made a two month comparison of model simulations (ZTDs, ZTDs + TGs, and TGs assimilation runs) with respect to independent radiosondes which were not assimilated into the model in our previous article Thundathil et al. (2024).

- Even though it is not the central topic of the study, it would be interesting to have details on the impact of gradients on the humidity field according to vertical levels. Where are they most informative? Perhaps adding the curve for the ZTD_1.0° experiment in Figure 5 would help?

Thank you for the comment. As per the suggestion of the reviewer, we have now replaced the plot with a new figure with all the experiment profiles as below:

[Figure]

**Figure 5.** The RMSE of specific humidity profiles compared to ERA5 for Control run (black), ZTD_0.5° run (blue), ZTD_1.0° run (green), ZTDGRA_1.0° run (red), and ZTDGRA_0.5° run (purple). Profiles were compared at five selected stations for 220 DA cycles, totaling 1100 profiles for the average plot.

- Do you think ZTD and gradients would have had the same impact if satellite observations had been assimilated? The same question applies to radar data.

Thank you for the question. We expect a positive impact with the assimilation of GNSS ZTDs and TGs if satellite and radar data are assimilated. However we expect the scale of improvement to be less. The quantification of the relative improvement with satellite and radar data would be a topic for future research.

---

## Author Response (AR2)

Dear Editor,

As per the comments put forth by Reviewer #3 and the editor, we have made the necessary changes to the manuscript. Here are the point-by-point responses to the comments. We take this opportunity to thank the reviewers and the editor for their valuable feedback, which helped us improve the manuscript and publish the results with all necessary inputs.

Best regards,
Rohith Thundathil

**Comments:**
* line 91: change to "we aim to further investigate"
Thank you for the correction. This has now been changed.
Line 91:
"In this study, we aim to further investigate the potential of TGs through a sensitivity experiment."
* lines 125 and 146: the citation should be "Dousa et al., 2016" (drop the first name Jan).
Done. Please refer to lines 125 and 146. "Douša et al., 2016"
* line 150: "to comply" instead of "to complying"
Corrected.
* line 155-156: drop the sentence "In order to compare the simulations with respect to independent GNSS observations, we removed excluded 18 stations from our dataset for validation purposes": this information is provided in the forthcoming sentences and is therefore repetitive.
Sentence removed. Now the line 156 reads "In line with the approach of Thundathil et al. (2024), we intentionally excluded 18 stations in Germany from our dataset for validation purposes."
* line 182: "squared" instead of "square" (I think)
Since it is a matrix type and not raised to a power, it should be called a square matrix. No changes have been made.
* line 193: "six-hourly" instead of "Six-hourly"
Corrected. Thank you.
* line 195: "input" instead of "inputs"?
Thank you. Done.
* line 200: as mentioned by referee #3: the RUC has not explained before, as it has been removed from an earlier version of the manuscript.
RUC has been replaced by six-hourly DA cycle. Please refer to line 199.
* line 214: "excluded stationS"
Thank you. Corrected.
* line 260: has A large error
Done.
* line 264: drop the , after TGs
Done.
* line 321: drop total before number (now it reads: "the total number of profiles totaled 1100", with is rather awkward).
Done.
* line 326: replace "she/he" with "one"
Done.
* line 365: where is Table A1? And replace "will summarize" with "summarizes"
Thank you for pointing it out. Table A1 has been replaced by Table 1. Please refer to line 364.